# MTD-Diorama: Moving Target Defense Visualization Engine for Systematic Cybersecurity Strategy Orchestration

**DOI:** 10.3390/s24134369

**Published:** 2024-07-05

**Authors:** Se-Han Lee, Kyungshin Kim, Youngsoo Kim, Ki-Woong Park

**Affiliations:** 1SysCore Lab., Convergence Engineering for Intelligent Drone, Sejong University, Seoul 05006, Republic of Korea; sehanlee141@gmail.com; 2Department of Computer and Information Security, Sejong University, Seoul 05006, Republic of Korea; 3Agency of Defense Development (ADD), Daejeon 34186, Republic of Korea; updatekim@add.re.kr; 4Electronics and Telecommunications Research Institute (ETRI), Daejeon 34129, Republic of Korea; blitzkrieg@etri.re.kr

**Keywords:** moving target defense, cyberattack surface, data visualization, classification

## Abstract

With the advancement in information and communication technology, modern society has relied on various computing systems in areas closely related to human life. However, cyberattacks are also becoming more diverse and intelligent, with personal information and human lives being threatened. The moving target defense (MTD) strategy was designed to protect mission-critical systems from cyberattacks. The MTD strategy shifted the paradigm from passive to active system defense. However, there is a lack of indicators that can be used as a reference when deriving general system components, making it difficult to configure a systematic MTD strategy. Additionally, even when selecting system components, a method to confirm whether the systematic components are selected to respond to actual cyberattacks is needed. Therefore, in this study, we surveyed and analyzed existing cyberattack information and MTD strategy research results to configure a component dataset. Next, we found the correlation between the cyberattack information and MTD strategy component datasets and used this to design and implement the *MTD-Diorama* data visualization engine to configure a systematic MTD strategy. Through this, researchers can conveniently identify the attack surface contained in cyberattack information and the MTD strategies that can respond to each attack surface. Furthermore, it will allow researchers to configure more systematic MTD strategies that can be used universally without being limited to specific computing systems.

## 1. Introduction

Today, with the advancements in information and communication technology (ICT), modern society is developing into a hyperconnected society in which various things are connected through the Internet [1,2]. The Internet of Things (IoT) is currently being used in various industrial fields, such as smart medical devices, autonomous vehicles, smart factories, and smart cities [2,3,4].

However, cyberattacks are also evolving with the development of ICT [2,3,4,5,6,7,8] and go beyond the simple threat of personal information leakage to significantly impacting human life and urban infrastructure (e.g., in-body medical devices and nuclear facility systems). A representative example is the IoT attack using the Mirai Botnet [9], where a vulnerable IoT device infected with malware took control of the host system connected to the IoT device and used it for a large-scale Denial-of-Service (DoS) attack, paralyzing the IoT service. In another case, an attack caused IoT devices to stop functioning or malfunction, paralyzing the smart network infrastructure and causing casualties [10,11].

A moving target defense (MTD) strategy was designed to respond to the evolving cyberattacks. The MTD strategy provides proactive actions that target mission-critical systems, and many research results are currently emerging [12,13]. However, there is a lack of indicators that suggest which system components can be used, making it difficult to configure a systematic MTD strategy. Additionally, there is a need to use visualization data as indicators to decide from an existing MTD strategy when a specific cyberattack occurs.

Accordingly, in this study, we surveyed and analyzed various existing MTD strategy results to derive critical components from three key perspectives [13]. Additionally, we used the Open Indicator of Compromise (OpenIOC) framework [14] to derive components for cyberattack information and build a dataset using the derived components. Next, we designed and implemented a component data visualization engine to provide visual information connecting the MTD strategy and cyberattack information components. The proposed data visualization engine can help existing MTD strategy researchers confirm whether the currently studied MTD strategy can respond to an actual cyberattack. In addition, it provides future researchers with various component combination indicators to configure new MTD strategies through component connection indicators between the existing MTD strategy research results and actual cyberattack information.

This study is an extension of our previous study [15] that focused on designing and implementing a data visualization engine. In the present study, we show that various components in a general computing system structure can be identified through the implemented data visualization engine, and connection points between existing MTD strategies and cyberattack information components can be identified. Additionally, we show that the proposed data visualization engine can be continuously used to manage and utilize MTD strategy information by constructing a new MTD strategy and immediately adding it to a data visualization engine.

The novelty of the proposed data visualization engine, named *MTD-Diorama*, is evident in two aspects.

First, *MTD-Diorama* effectively visualizes MTD strategies, enabling mission-critical system administrators and cybersecurity experts to respond to various cyberattack scenarios. Unlike traditional MTD strategies that primarily focus on predicting and mitigating attacks, *MTD-Diorama* enhances the visual representation and interactivity of these strategies, facilitating a more intuitive understanding and rapid response.

Second, *MTD-Diorama* integrates multidimensional security data analysis and visualization to help briefly understand the various threat factors that can occur in a complex security environment. Beyond being a simple data visualization tool, it provides a way for cybersecurity practitioners or cybersecurity researchers to evaluate and optimize the effectiveness of MTD strategies in real time.

The remainder of this paper is organized as follows. Section 2 provides an overview of the MTD strategy and describes OpenIOC, which was used to derive the cyberattack information components. Section 3 describes the derivation and classification of the components for the MTD strategy and cyberattack information, as well as the dataset configuration. Section 4 describes the design and implementation of the *MTD-Diorama* data visualization engine, while Section 5 describes its use. Finally, Section 6 presents the conclusions and directions for future research.

## 2. Preliminary Background

### 2.1. Moving Target Defense Strategy

A moving target defense (MTD) is a systematic protection strategy that actively and continuously changes the attack surface targeted within a mission-critical system [12]. Thus, the attack surface exposed to the attacker appears chaotic, and the vulnerabilities discovered in advance by the attacker can be nullified over time. An overview of the MTD strategy is shown in Figure 1.

Figure 1 illustrates the workflow for the concept of an MTD strategy within a mission-critical system. Each step involves detecting an attack, responding to it, and reconfiguring the system. The following is a detailed explanation of the workflow:1.Attack Detection and Alert-A malicious user attempts an attack on the system, targeting Conf_03.-When the attack reaches Conf_03, the MTD Module within the system detects it.-Conf_03 sends an attack notification to the MTD Module (① Alert) to notify that an attack has occurred.-Simultaneously, in the ② Detect phase, the module identifies the nature and location of the attack.2.Response and Control-The MTD Module issues a ③ Control command to respond to the attack.-This control command instructs the system to change its configuration.3.System Reconfiguration-After the attack is detected, the system reconfigures itself by replacing the compromised configuration Conf_03 with another configuration Conf_12.-This change can affect other parts of the system, ensuring that Conf_03 is replaced with Conf_12 to respond to the attack.4.System Reoperation-After the reconfiguration, the system operates with the new configuration Conf_12.-Other configurations such as Conf_01, Conf_02, Conf_04, Conf_05, Conf_06, and Conf_n remain unchanged, while Conf_03 has now been replaced by Conf_12.

MTD strategy can reduce the likelihood of successful attacks and effectively improve the resilience and security of mission-critical systems. Moreover, the MTD strategy changes the system security paradigm from the existing passive form of defense to an active defense against cyberattacks [16,17,18,19,20,21,22].

This strategy actively changes the mission-critical system components (attack surfaces possibly subjected to cyberattacks). Thus, the effect of obfuscating system components can be achieved such that attackers cannot analyze the system. It also reduces attack opportunities and requires attackers to invest more time in analyzing mission-critical systems.

### 2.2. Three Perspectives of the MTD Strategy

The MTD strategy can be classified from three perspectives [12,19,22]: What, When, and How to move. The components that make up each perspective come together to form one MTD strategy and determine the direction of cybersecurity technology development using the MTD strategy.

The following is an explanation of each perspective:What to move—This perspective concerns which components (the attack surface that an attacker can identify) of the mission-critical system should be moved or mutated when implementing security technology using the MTD strategy. The attack surface defined here incorporates one or more components that comprise the operating system, hardware, and software subject to cyberattacks or containing vulnerabilities, such as IP addresses, MAC addresses, and port numbers in the network area.When to move—This time-series perspective determines when to move or mutate the mission-critical system components (the attack surface an attacker can identify). This perspective can affect performance (or availability) when implementing the MTD strategy technology in mission-critical systems. If the frequency of the protection process corresponding to that aspect within the protection technology is too low, the likelihood that an attacker attacks quickly increases. In contrast, if the frequency is too high, although a high level of security service can be provided to the protected system, the resulting large overhead can deteriorate the mission-critical system’s performance and service availability.How to move—This perspective determines how to move or mutate mission-critical system components. The two tasks that are performed to achieve this goal are selection and replacement. The selection operation selects a new component based on the available data using various methods, such as random data selection or assigning new data to a previously moved or mutated component. The replacement operation transforms one or more components selected through a protection technology process into a new component or exchanges data with one or more components in a mission-critical system.

### 2.3. Open Indicators of Compromise (OpenIOC) Framework

The OpenIOC framework is an open-source-based cybersecurity incident indicator framework developed by Mandiant [23,24,25,26]. It provides various indices to identify data from different attack surfaces contained in a single piece of cyberattack information [24,25]. This incident indicator is widely used when analyzing cyberattacks within governments or companies and is provided as extensible markup language (XML) documents that help capture various artifacts about specific cyberattack information. It is highly recognized when sharing information on cyber incidents and has excellent interoperability with signature-based security devices such as intrusion prevention or detection systems [23,24].

### 2.4. Cyberattacks in IoT Systems

IoT systems are evolving to enable their use in various fields by the addition of real-time networking characteristics to the structure of existing computing systems [1]. As their usability becomes more important, the frequency of cyberattacks targeting IoT systems is gradually increasing [2,3,4,5,6,7,8,9,10,11].

An example of cyberattacks targeting IoT systems includes Reverse Engineering to analyze device firmware vulnerabilities in IoT systems, Man-in-the-Middle attack, Sniffing, Spoofing, and Replay attacks to attack network communication between IoT devices, and Denial-of-Service (DoS) and Side-Channel attacks to attack services operated by IoT systems [27]. From a game theory perspective, these attacks on IoT systems and the defensive actions to stop them can be viewed as a competition to maximize the rewards (reward for attack, reward for defense) from each perspective of the attacker and the defender [28].

Zhang et al. [29] introduced an anti-jamming scheme based on a Colonel Blotto Game to prevent jamming attacks on underwater acoustic backscatter communication used in gliding autonomous underwater vehicles (AUVs). In their study, they modeled the competition between two players, an attacker and a defender, for a limited resource budget to analyze the competitive interaction between surface sink nodes (SNs) and AUVs.

Pirozmand et al. [30] introduce a method that applies game theory to develop an effective intrusion detection system performance in a cloud-fog-based IoT network environment. In their study, the operation of the intrusion detection system was structured as a dynamic game between two players, an attacker and a defender, and a non-participatory dynamic game, and the parameters for attack and defense were extracted and analyzed.

Abdalzaher et al. [31] introduced a game theory approach to enhance system security and data trustworthiness in wireless sensor network (WSN)-based IoT environments. In their study, a repeated game model between two players, an attacker and a defender, was proposed to enhance clustered WSN-based IoT security and data trustworthiness.

Hence, as the complexity, efficiency, and usability of IoT systems increases, the rewards from both attacker and defender perspectives are also maximized. Based on this, cyberattacks on IoT systems are becoming increasingly intelligent, and it is time for an active defense strategy to be developed for IoT systems.

## 3. Component Classification and Dataset Configuration

This section describes the derivation and classification of the MTD strategy and cyberattack information components for configuring a visualization dataset for use in a data visualization engine. We then describe how to configure a visualization dataset using the derived components.

### 3.1. Component Classification of the MTD Strategy

To derive the components of the MTD strategy, we first surveyed and analyzed various MTD strategy research results. The results are given in Table 1.

The current MTD strategy research proposes protection algorithms and systems based on software. In addition, many studies focus on protecting components (e.g., IP address, MAC address, port number, and protocol) identified in the network area. In addition to software-based technologies, research has focused on hardware-based technologies. An example is in [36], which reconfigured the controller area network (CAN) communication bus circuit inside an autonomous vehicle using a switch and proposed a network flow control mechanism to transmit and receive authenticated CAN network packets.

Based on Table 1, the components that comprise the three perspectives of the MTD strategy were derived and classified. The components are given in Table 2.

First, from the perspective of “What to move”, 10 components were derived. The components include attack surfaces that can be identified when an attacker performs a cyberattack targeting a mission-critical system. Second, from the perspective of “When to move”, two components that were guaranteed to contain a single piece of time-series information when performing the protection process using the MTD strategy were derived. Finally, six components were derived from the perspective of “How to move”. The components exhibit a characteristic that changes the attack surfaces an attacker can identify while performing the protection process using an MTD strategy.

### 3.2. Component Classification of Cyberattack Information Using the OpenIOC Framework

This study used OpenIOC to derive the components (attack surfaces) of the cyberattack information. It was confirmed that various artifacts of a single attack’s information expressed through OpenIOC represented the attack surface from the perspective of “What to move”. As an example, the classification of components for Stuxnet attack information is shown in Figure 2.

### 3.3. Configuration of the Dataset for the Data Visualization Engine

From the previously derived MTD strategies and cyberattack information components, we identified the attack surface as the common component. Therefore, the proposed data visualization engine uses attack surface components to express the correlation between the MTD strategy and cyberattack information. In addition, we aimed to provide visual information regarding existing MTD strategies that could be used for single-attack information.

To this end, the need to express the components of the attack surface according to the general system configuration was confirmed, and various sections comprising one system and the attack surface corresponding to each section were classified. For deriving and classifying components of cyberattack information, we identified five system sections (Network, Storage, OS Configuration, Application, and Log Sections) by analyzing various artifact indexes provided by the OpenIOC framework [25].

The method for configuring the MTD strategy and cyberattack information datasets is shown in Figure 3. And an example of the dataset configuration based on the previously derived component classification results is shown in Figure 4.

The section codes corresponding to each section of a general computing system are generated based on the section classification method of the attack surface component. We store the data in the Section Code attribute commonly configured in the two datasets. In addition, to classify the attack surface corresponding to each section, a dataset is configured by classifying the “What to move” component of the MTD strategy and the components of the cyberattack information derived and classified through the keywords for each section.

The explanation of Figure 4 is as follows. The component derivation results are commonly loaded through the Component Loader module when configuring the MTD strategy component and cyberattack information datasets. Subsequently, the attack surface components are separated and added to the dataset, along with the remaining components. Attack surface components are added to the dataset by classifying the system section to which each attack surface component corresponds according to the system section classification method shown in Figure 3.

An example of the MTD strategy component dataset configured using the process shown in Figure 4 is presented in Table 3, and an example of the cyberattack information component dataset is presented in Table 4.

## 4. Design and Implementation of the Data Visualization Engine

In Section 3, based on the research results of the MTD strategy, we derived the MTD strategy components and constructed a dataset. We also constructed a dataset on cyberattack information. However, there are limitations to simply using a dataset to show the connections between two datasets. First, the dataset itself is simply a collection of data, so it is difficult to directly utilize it for other research. Second, even if it is claimed that there is a connection between the MTD strategy and cyberattack information, it is difficult to utilize it easily if quantitative information (graphs, probabilities, etc.) is not provided.

In this study, we derived and classified the components of existing MTD strategies and cyberattack information and provided visual information to understand the connection between the two. In addition, a component data visualization engine was designed to provide visual information to identify the MTD strategy that could be utilized among the various attack surfaces in actual cyberattacks.

The environment for implementing *MTD-Diorama* and the environment for running the implemented engine are shown in Table 5.

The system design of the proposed *MTD-Diorama* data visualization engine is illustrated in Figure 5.

The system structure designed to implement the data visualization engine is as follows. The Dataset Loader module loads the previously configured MTD strategy and cyberattack information component datasets. The Dataset Handler module creates a new dataset for the data visualization engine. The Attack Information Selector module identifies attack information selected by the engine user. Finally, the Attack Information & MTD Viewer, Attack Surface Analyzer, and Attack Surface & MTD Viewer modules generate and provide various types of visual information based on the attack information the user selects. The top of Figure 5 shows the final implemented data visualization engine, which is described as the following:①The computing system section classified along with data construction in Section 3 is expressed. When the user selects the desired cyberattack information and runs the engine, a computing system attack surface associated with the chosen cyberattack information appears and an MTD strategy that can respond to each attack surface is expressed.②The MTD strategy component dataset that can respond to the cyberattack information selected by the user is displayed in a table format. Users can check which components of an MTD strategy can respond to each piece of cyberattack information and use it as an indicator when constructing a new MTD strategy.③The chart provides visual information to check which section of each computing system the cyberattack information selected by the user is attacking and the attack surface in that section.④The chart identifies MTD strategy information that can respond to the cyberattack information selected by the user and provides visual information on the rate at which the identified MTD strategy information can respond to the cyberattack information.

## 5. Utilization of a Data Visualization Engine

The *MTD-Diorama* visually shows the connectivity of existing MTD strategy information and the components of cyberattack information using the OpenIOC framework. In addition, it helps formulate an MTD strategy to respond systematically to various cyberattacks and can be used as an indicator to determine the configuration direction of a new MTD strategy. Figure 6 shows an example when running *MTD-Diorama*.

The explanation for Figure 6 is as follows. There are four attack surfaces (IP Address, MAC Address, File Path, and Process PID) in the selected Attack_01 information, and there are six MTD strategies (MTD_01, MTD_02, MTD_03, MTD_04, MTD_06, and MTD_09) that can respond to them. In this case, MTD_01 and MTD_02 can effectively respond to Attack_01 each with a 50% probability.

The results shown in Figure 6 were obtained from an engine run using the datasets listed in Table 3 and Table 4. As a data visualization engine, *MTD-Diorama* provides various visual information and can easily confirm connection points between the MTD strategy and cyberattack information. Therefore, existing MTD strategy researchers can use it as an indicator of the practicality of their studied MTD strategy. The results when selecting different attack information are shown in Figure 7 and Figure 8, and the accuracy and usability of these results increase depending on the size of the datasets.

In addition, the *MTD-Diorama* can be used to configure new MTD strategies by combining the components of existing MTD strategy research results, as shown in Figure 9.

Moreover, the visual information displayed in *MTD-Diorama* can be used as an indicator to determine the research or development direction when configuring new MTD strategies.

When a new MTD strategy or cyberattack information is identified, the data visualization engine can immediately add it to the dataset and utilize it. An example is shown in Figure 10. Even if new information is identified, users can add the desired information at any time based on the dataset configuration described in Section 3. The added data are immediately reflected in the data visualization engine. Accordingly, users can continuously use it as an indicator to check the connection between the existing and new MTD strategy and cyberattack information.

As can be seen from the research results of the MTD strategy surveyed and analyzed in Section 3.1, the field in which the MTD strategy has most widely been used to date is the network field. Using *MTD-Diorama* as a visualization engine in a network field, especially in a software-defined network (SDN) environment [42,43,44], can suggest new strategic possibilities for network security. By combining the visualization and component dataset-understanding capabilities of *MTD-Diorama* with the central control and flexible network configuration capabilities of SDN, a more efficient and effective cybersecurity defense system can be built. This provides great advantages in implementing dynamic defense strategies and responding to real-time threats, especially in complex network environments. 

## 6. Conclusions

With the development of ICT, cyberattacks are also developing and increasing. The MTD strategy helps configure a preemptive defense strategy for a mission-critical system and respond to these cyberattacks. However, with the increasing diversity of computing systems, there is a need to formulate a systematic MTD strategy that can be utilized from the perspective of a general computing system. Therefore, an indicator that can be used as a reference is required.

This study proposes a data visualization engine visually demonstrating the connection between the MTD strategy and cyberattack information. Using the proposed engine, users can check the components of the MTD strategy and cyberattack information and use them as indicators when configuring a new security strategy.

However, there are shortcomings in directly proving how systematic the newly configured MTD strategy using *MTD-Diorama* is. To achieve this, when a newly configured MTD strategy technology operates in a specific system, a method is needed to measure the overhead of that system and show the actual internal system configuration. In addition, the visualization method using the current component dataset has a shortcoming in that it is difficult to derive results related to the status information of the actual computing system. To achieve this, a more detailed component classification method is needed, such as the usage data range of each component, data type, actual data values, etc.

In the future, we plan to study an extended and specified component classification scheme and develop an advanced *MTD-Diorama* data visualization engine based on it to show how not only the components in a large-scale concept but also the detailed data of the components interact in a computing system. Moreover, we plan to study ways to identify what happens within a mission-critical system in real time when responding to a cyberattack by simulating how the MTD strategy operates in a computing system.

Finally, we plan to build a testbed environment based on digital twins [45] for simulation verification of the advanced *MTD-Diorama*. This will be used to compare and analyze the real-time cyberattack response results of the MTD strategy in a virtually implemented computing system with the response results in an actual computing system and derive an interactive relationship. To achieve this, we plan to build, test, and verify this using the Software-In-The-Loop Simulation (SITL) method and the Hardware-In-The-Loop Simulation (HILS) method [46,47].

## Figures and Tables

**Figure 1 sensors-24-04369-f001:**
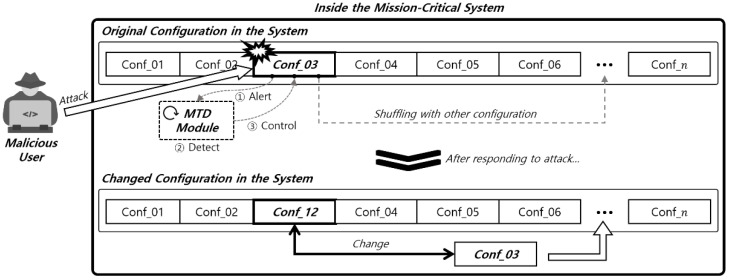
An overview of the moving target defense strategy.

**Figure 2 sensors-24-04369-f002:**
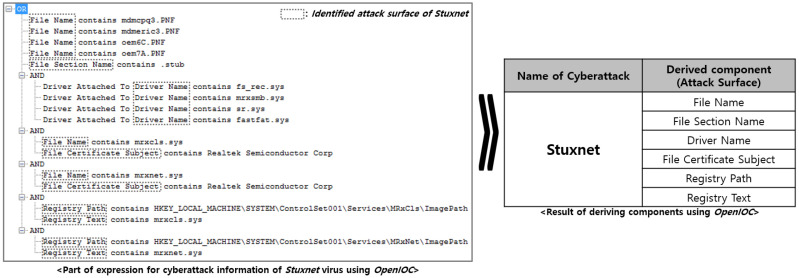
A result of deriving components for Stuxnet virus information.

**Figure 3 sensors-24-04369-f003:**
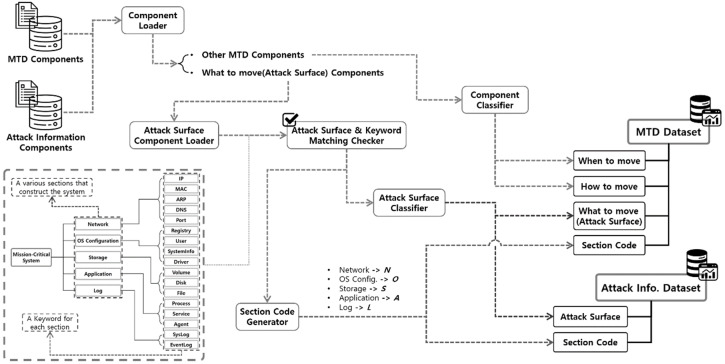
Design diagram for dataset configuration.

**Figure 4 sensors-24-04369-f004:**
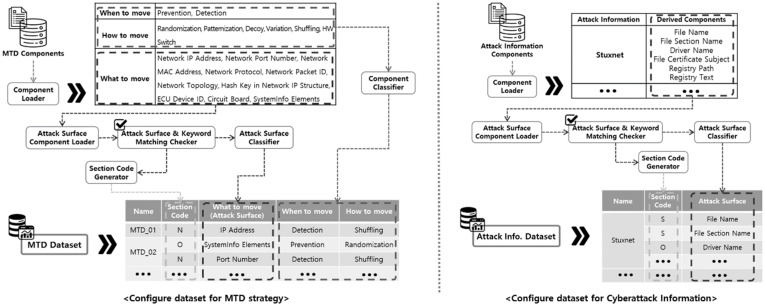
Process of configuring the MTD strategy and cyberattack information datasets.

**Figure 5 sensors-24-04369-f005:**
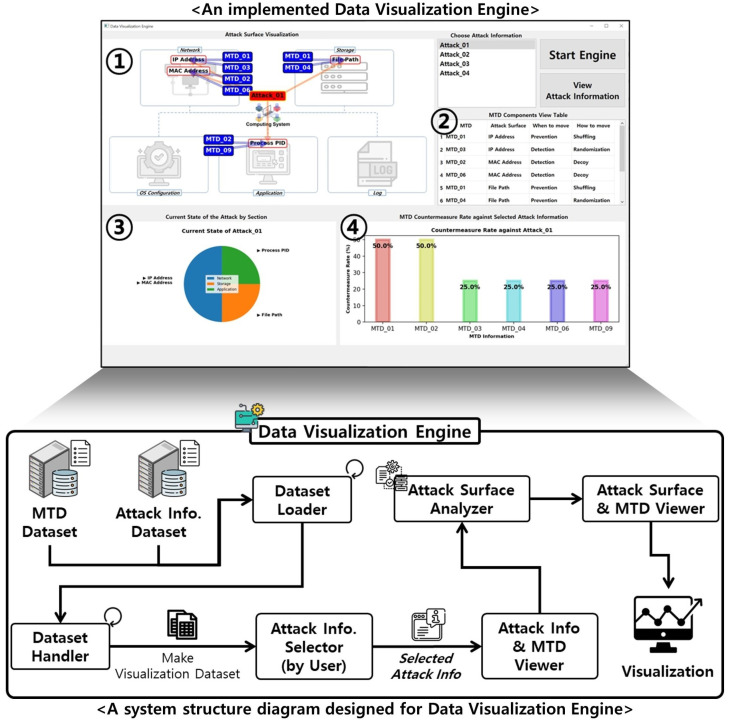
System design and implementation result for *MTD-Diorama*.

**Figure 6 sensors-24-04369-f006:**
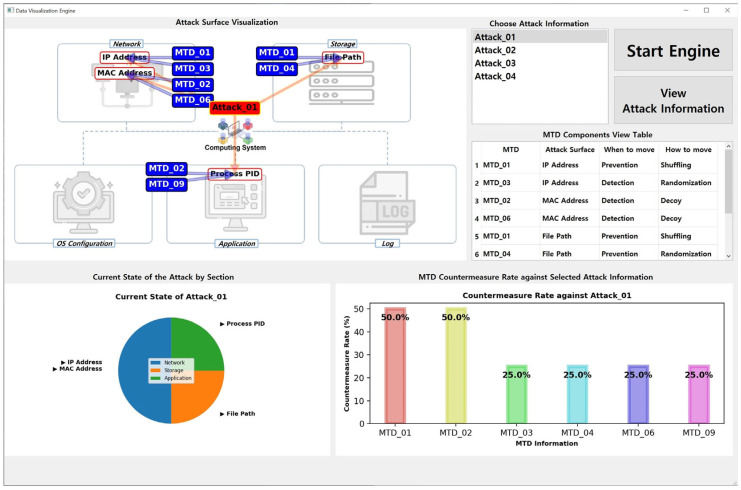
Result of the *MTD-Diorama* operation (when choosing Attack_01).

**Figure 7 sensors-24-04369-f007:**
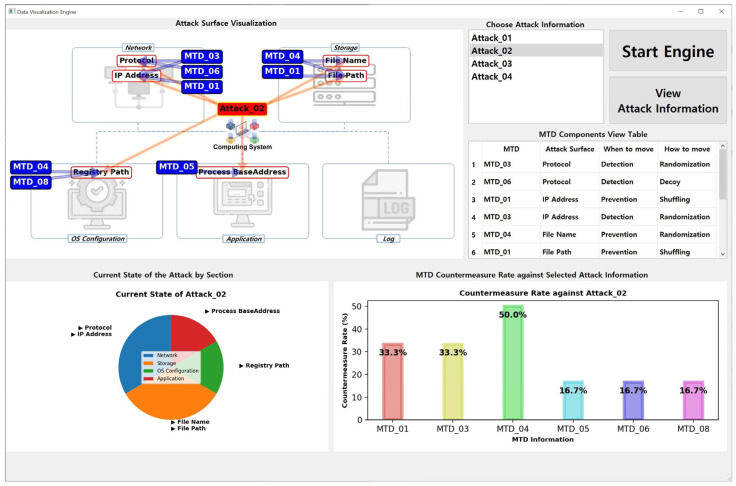
Result of the *MTD-Diorama* operation (when choosing Attack_02).

**Figure 8 sensors-24-04369-f008:**
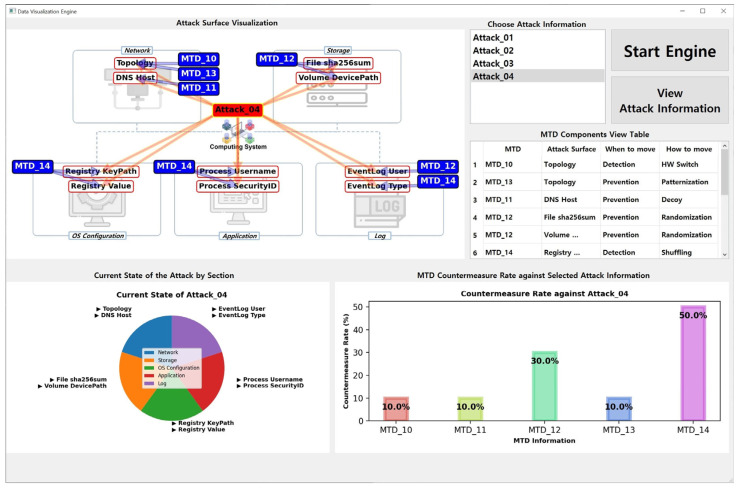
Result of the *MTD-Diorama* operation (when choosing Attack_04).

**Figure 9 sensors-24-04369-f009:**
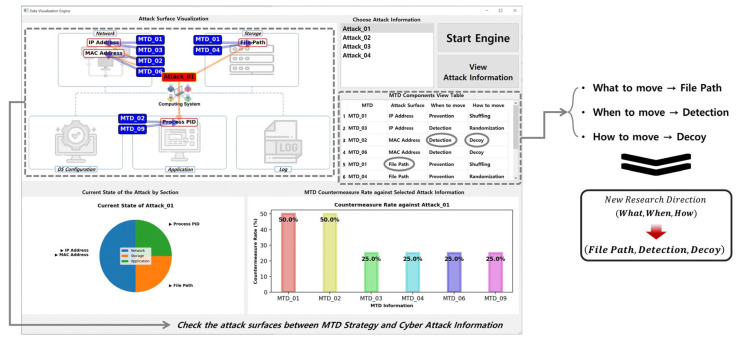
Method of deriving a new MTD strategy research direction using *MTD-Diorama*.

**Figure 10 sensors-24-04369-f010:**
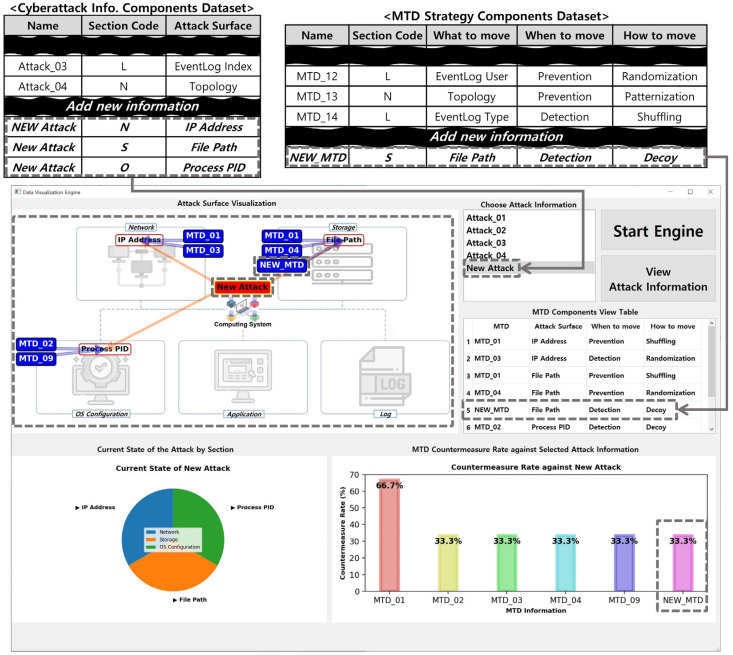
Example of *MTD-Diorama* when adding new information in the dataset.

**Table 1 sensors-24-04369-t001:** Analysis of existing MTD strategy research results.

Author	Summary	Targeted System Component
Moon [32]	Block the continuity of advanced persistent threat (APT) attacks by deriving system environment elements vulnerable to APT attacks.	System Environment Elements
Leem et al. [33]	Use an attack target disruption mechanism based on a preposition hash table (PHT) to reduce the risk of exposure of hash values in communication network packets to identify unmanned flying objects (drones).	Hash Key in Network IP Structure
Park et al. [34]	Mutate the IP address and port number according to the network-based MTD strategy called hidden tunnel networking.	Network IP Address, Network Port Number
Hong et al. [35]	Provide optimal network configuration through an SDN network topology analysis using shuffle-based online MTD.	Network Topology
Narantuya et al. [36]	Shuffle IP addresses using multiple software-defined network (SDN) controllers in SDN-based network environments.	Network IP Address
Woo et al. [37]	Shuffle the controller area network (CAN) IDs to protect the in-vehicle network using network address shuffling.	CAN Network ID
Brown et al. [38]	Provide a new CAN bus protocol that uses randomization seeds to generate ECU IDs randomly.	ECU ID on the CAN Bus
Park et al. [39]	Use the network protocol variation patterns to ensure that only users who know the patterns can access the server.	Network Protocol
Yoon et al. [40]	Shuffle the network configuration properties (e.g., MAC address, IP address, port number) based on an attack graph of the host system to be protected.	Network Configuration Properties
Groza et al. [41]	Use a strategy to secure CAN network communications by configuring switches in the CAN bus circuit inside the vehicle.	CAN Bus Circuit Board

**Table 2 sensors-24-04369-t002:** Critical components derived based on three perspectives of the MTD strategy.

Perspective	Derived Components
When to move	Prevention, Detection
What to move (Attack Surface)	Network IP Address, Network Port Number, Network MAC Address, Network Protocol, Network Packet ID, Network Topology, Hash Key in Network IP Structure, ECU Device ID, Circuit Board, System Information Elements
How to move	Randomization, Patternization, Decoy, Variation, Shuffling, Hardware Switch

**Table 3 sensors-24-04369-t003:** Example of the MTD strategy component dataset.

Name	Section Code	What to Move (Attack Surface)	When to Move	How to Move
MTD_01	N	IP Address	Prevention	Shuffling
MTD_01	S	File Path	Prevention	Shuffling
MTD_02	N	MAC Address	Detection	Decoy
MTD_02	A	Process PID	Detection	Decoy
MTD_03	N	Protocol	Detection	Randomization
MTD_03	N	IP Address	Detection	Randomization
MTD_04	S	File Name	Prevention	Randomization
MTD_04	S	File Path	Prevention	Randomization
MTD_04	O	Registry Path	Prevention	Randomization
MTD_05	A	Process BaseAddress	Prevention	Variation
MTD_06	N	IP Hash Key	Detection	Decoy
MTD_06	N	MAC Address	Detection	Decoy
MTD_06	N	Protocol	Detection	Decoy
MTD_06	L	EventLog EID	Detection	Decoy
MTD_07	S	File Type	Prevention	Variation
MTD_07	S	File Timestamp	Prevention	Variation
MTD_07	L	EventLog Index	Prevention	Variation
MTD_08	S	File SectionName	Detection	Shuffling
MTD_08	O	Registry Path	Detection	Shuffling
MTD_08	O	Registry Text	Detection	Shuffling
MTD_09	A	Process PID	Prevention	Shuffling
MTD_09	A	Process Timestamp	Prevention	Shuffling
MTD_10	N	Topology	Detection	HW Switch
MTD_11	N	DNS Host	Prevention	Decoy
MTD_12	S	File sha256sum	Prevention	Randomization
MTD_12	S	Volume DevicePath	Prevention	Randomization
MTD_12	L	EventLog User	Prevention	Randomization
MTD_13	N	Topology	Prevention	Patternization
MTD_14	L	EventLog Type	Detection	Shuffling

**Table 4 sensors-24-04369-t004:** Example of the cyberattack information component dataset.

Name	Section Code	Attack Surface
Attack_01	N	IP Address
Attack_01	N	MAC Address
Attack_01	S	File Path
Attack_01	A	Process PID
Attack_02	N	Protocol
Attack_02	N	IP Address
Attack_02	S	File Name
Attack_02	S	File Path
Attack_02	O	Registry Path
Attack_02	A	Process BaseAddress
Attack_03	N	IP Hash Key
Attack_03	N	MAC Address
Attack_03	N	Protocol
Attack_03	S	File Type
Attack_03	S	File Timestamp
Attack_03	S	File SectionName
Attack_03	O	Registry Path
Attack_03	O	Registry Text
Attack_03	A	Process PID
Attack_03	A	Process Timestamp
Attack_03	L	EventLog EID
Attack_03	L	EventLog Index
Attack_04	N	Topology
Attack_04	N	DNS Host
Attack_04	S	File sha256sum
Attack_04	S	Volume DevicePath
Attack_04	O	Registry KeyPath
Attack_04	O	Registry Value
Attack_04	A	Process Username
Attack_04	A	Process SecurityID
Attack_04	L	EventLog User
Attack_04	L	EventLog Type

**Table 5 sensors-24-04369-t005:** The implementation and execution environment of the *MTD-Diorama*.

Type	Environment
Implementation	OS: Windows 10 CPU: Intel Core i5-8500 3.0 GHz Language: Python v3.10 Library: PyQT v5 (for GUI)
Execution	OS: Windows 10 CPU: Intel Core i5-8500 3.0 GHz Memory: 8 GB

## Data Availability

All data are contained within this article.

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
