# Peer review of "MTD-Diorama: Moving Target Defense Visualization Engine for Systematic Cybersecurity Strategy Orchestration"

_sensors, 2024, doi:10.3390/s24134369_

Round 1

Reviewer 1 Report

Comments and Suggestions for Authors

1. What is the main question addressed by the research?

The moving target defense (MTD) strategy was designed to protect mission-critical systems from cyberattacks. The MTD strategy shifted the paradigm from passive to active system defense. However, there is a lack of indicators that can be used as a reference when deriving general system components, making it difficult to configure a systematic MTD strategy.

2. Do you consider the topic original or relevant in the field? Does it address a specific gap in the field?

Yes. The problem solved in this paper is relevant to this journal.

3. What does it add to the subject area compared with other published material?

The main contributions of this paper include:

- Survey and analyze existing cyberattack information and MTD strategy research results to configure a component dataset.

- Find the correlation between the cyberattack information and MTD strategy component datasets and use this to design and implement the MTD-Diorama data visualization engine to configure a systematic MTD strategy.

4. What specific improvements should the authors consider regarding the methodology? What further controls should be considered?

- The novelty of this paper should be clearly presented in the introduction part.

- The implementation and execution environment of the data visualization engine should be provided in the paper.

- How to combine the authors’ work with these works in terms of SDN, such as B-DNS: a secure and efficient DNS based on the blockchain technology, Detection and mitigation of DoS attacks in software defined networks, Flooddefender: protecting data and control plane resources under SDN-aimed DoS attacks. More discussion should be added in the paper.

- Some potential research directions and future works could be discussed in the end of this paper.

5. Are the conclusions consistent with the evidence and arguments presented and do they address the main question posed?

Yes.

6. Please include any additional comments on the tables and figures.

- Tables and figures are easy to understand.

Author Response

Thank you for your valuable comments.
We have revised the paper to accommodate all of your comments and have written a response to them.
Please see the attachment.
Thank you.

Reviewer 2 Report

Comments and Suggestions for Authors

1. The detailed discussion on the MTD strategy framework in Fig. 1 can be well provided. The workflow needs to be elaborated.

2. The authors are suggested to further discuss the background of cyber-attacks in IoT systems by considering the game theoretical methods used to deal with the cyber-attacks, such as “Anti-Jamming Colonel Blotto Game for Underwater Acoustic Backscatter Communication,” IEEE Transactions on Vehicular Technology, DOI: 10.1109/TVT.2024.3367935.

3. The motivation of designing the MTD-Diorama data visualization engine should be discussed.

4. The effectiveness of the proposed data visualization engine should be verified through necessary simulation experiments.

5. As the future work, the authors are suggested to consider employing the digital twins to assist the system design, such as “Digital Twin-Assisted Edge Computation Offloading in Industrial Internet of Things With NOMA,” IEEE Transactions on Vehicular Technology, September 2023.

Comments on the Quality of English Language

None.

Author Response

(The authors gave the same response as above.)

Round 2

Reviewer 1 Report

Comments and Suggestions for Authors

Most of the issues have been solved well. In Section 5, it's better to add some references about SDN for comment 3.

Author Response

(The authors gave the same response as above.)
